# Osseointegration of Sandblasted and Acid-Etched Implant Surfaces. A Histological and Histomorphometric Study in the Rabbit

**DOI:** 10.3390/ijms22168507

**Published:** 2021-08-07

**Authors:** Eugenio Velasco-Ortega, Iván Ortiz-Garcia, Alvaro Jiménez-Guerra, Enrique Núñez-Márquez, Jesús Moreno-Muñoz, José Luis Rondón-Romero, Daniel Cabanillas-Balsera, Javier Gil, Fernando Muñoz-Guzón, Loreto Monsalve-Guil

**Affiliations:** 1Comprehensive Dentistry for Adults and Gerodontology, Master in Implant Dentistry, Faculty of Dentistry, University of Seville, 41009 Seville, Spain; evelasco@us.es (E.V.-O.); ivanortizgarcia1000@hotmail.com (I.O.-G.); alopajanosas@hotmail.com (A.J.-G.); enrique_aracena@hotmail.com (E.N.-M.); je5us@hotmail.com (J.M.-M.); jolurr001@hotmail.com (J.L.R.-R.); danielcaba@gmail.com (D.C.-B.); mmonsalve2@us.es (L.M.-G.); 2Bioengineering Institute of Technology, Universitat Internacional de Catalunya, 08017 Barcelona, Spain; xavier.gil@uic.cat; 3Ibone Lab SL, Department of Veterinary Clinical Sciences, Faculty of Veterinary, University of Santiago de Compostela, 27002 Lugo, Spain

**Keywords:** osseointegration, implant surface, sandblasted and acid-etched surface

## Abstract

Titanium surface is an important factor in achieving osseointegration during the early wound healing of dental implants in alveolar bone. The purpose of this study was to evaluate sandblasted-etched surface implants to investigate the osseointegration. In the present study, we used two different types of sandblasted-etched surface implants, an SLA™ surface and a Nanoblast Plus™ surface. Roughness and chemical composition were evaluated by a white light interferometer microscope and X-ray photoelectron spectroscopy, respectively. The SLA™ surface exhibited the higher values (Ra 3.05 μm) of rugosity compared to the Nanoblast Plus™ surface (Ra 1.78 μm). Both types of implants were inserted in the femoral condyles of ten New Zealand white rabbits. After 12 weeks, histological and histomorphometric analysis was performed. All the implants were osseointegrated and no signs of infection were observed. Histomorphometric analysis revealed that the bone–implant contact % (BIC) ratio was similar around the SLA™ implants (63.74 ± 13.61) than around the Nanoblast Plus™ implants (62.83 ± 9.91). Both implant surfaces demonstrated a favorable bone response, confirming the relevance of the sandblasted-etched surface on implant osseointegration.

## 1. Introduction

Currently, dental implants are considered as a successful alternative to replace missing teeth with a predictable prognosis and are widely used in oral implantology for prosthodontic restorations [1,2,3,4]. Experimental studies in this field demonstrated that titanium is an excellent material due to its characteristics of biocompatibility, stability and resistance [5,6,7] achieving the osseointegration of the device. This phenomenon is characterized by the intimate interaction between the implant and bony tissues that results in a mechanical anchorage of the implant [8]. Osseointegration may be influenced by several factors such as the macroscopic design and the surface characteristics, the volume and density of bone at the implant site and the forces applied after the implant insertion [9,10].

Titanium implant surfaces play an important factor to achieve osseointegration because its physical and chemical properties are involved in the early wound healing after the surgical insertion in alveolar bone. In fact, results of cell adhesion, proliferation and differentiation from in vitro studies have suggested a positive correlation between surface roughness and cellular attachment with an increment in the osteoblast-like cell activity [11,12].

During the last decades, several methods have been developed to increase the roughness of the implants. These include titanium plasma spray, hydroxyapatite coating, acid etching, blasting and combinations of them [13,14]. All these methods have demonstrated that the rough surfaces are important for the long-term success of restorations with implants [15,16]. A firmer fixation is obtained with a rough surface than with a smooth one. Several studies have shown an increment in the bone implant contact compared with a machined surface, improving the quality of the interface and a stronger bone response [17,18,19]. An experimental study in animals investigated the early bone response of the titanium dental implants with different surface characteristics [19]. Several rough surfaces were compared with a machined surface. All the surface-modified implants showed a superior initial bone response compared with the control. Histological findings suggested that various surface treatments provided favorable bone responses for early functioning. The study concluded that surface modifications (i.e., blasting, anodic oxidation) showed faster osseointegration and bone healing compared with the turned titanium surface [19].

One of the most-used methods to increase the rugosity in implant dentistry is the sandblasted and acid-etched (SLA) surface treatment. An SLA surface is made by sandblasting the turned surface with large-grit particles (i.e., alumina) in a size range between 250 and 500 µm. Later, the surface is chemically treated using an acid [6,13], such as sulfuric, hydrochloric and nitric acid. Topographically, the SLA surface achieves a roughness with large dips, sharp edges and small micro pits to increase the contact surface as well as osseointegration [20,21]. The SLA surface has demonstrated good biological properties. Experimental designs reported that SLA increases the apposition of platelets and induces cell migration and differentiation. In fact, the SLA surface provides a favorable biological space for cell attachment, inducing the proliferation and growth of human osteoblasts [22,23,24].

In this paper, we present the results of a preclinical trial of two sandblasted and acid-etched implants made with a different chemical treatment. The comparison was made according to UNE-EN ISO 10993-6 in a rabbit model to investigate the bone response through histological and histomorphometric evaluation.

## 2. Results

### 2.1. Surface Characterization and Roughness

The characteristics of the new surface created were followed by scanning electronic microscopy (SEM) and Energy Dispersive X-ray Spectroscopy. SEM revealed differences between SLA and Nanoblast surfaces. SLA™ implants showed a rough profile characterized by irregular cavities alternated with metal peaks. The residuals of alumina used in the sandblasting process were still visible over the surface. In contrast to SLA, the Nanoblast Plus™ showed a homogeneous rough profile characterized by small cavities alternated with metal peaks (Figure 1 and Figure 2).

A roughness investigation underlined the differences between the two surfaces in terms of surface profile parameters. In particular, the SLA™ surface exhibited the higher values (Ra 3.05 μm) of rugosity compared to the Nanoblast Plus™ surface (Ra 1.78 μm). The parameter Rq, which expressed the irregularities distribution, was higher for the SLA™ surface compared to Nanoblast Plus™ surface (3.81 μm versus 2.23 μm). The parameter Rz, which expressed the differences between the highest and lowest points, was higher for the SLA™ surface compared to Nanoblast Plus™ (21.50 μm versus 14.11 μm). These data demonstrated a higher roughness of the surface in implants treated with SLA™ than Nanoblast Plus™ implants, but there was no significant difference between the two types of surface implants (*p >* 0.05) (Table 1).

### 2.2. Surface Composition

Chemical composition was performed by Energy Dispersive X-ray Spectroscopy. Titanium appeared as the main element of the alloy. The highest average percentage of titanium was for the Nanoblast Plus™ surface with 98.07% compared to the SLA™ surface, with 51.99%.

Common elements such as carbon was found in both surface implant systems. The lowest average percentage of carbon was for the Nanoblast Plus™ surface with 1.92% compared to the SLA™ surface, with 2%. Oxygen and aluminum were only found in the SLA™ surface, with 26.67% and 19.02%, respectively (Figure 3 and Figure 4).

### 2.3. Histological and Histomorhometric Study

One experimental animal (number 2) died during the healing phase. The remaining nine experimental animals healed without complications and at the time of sacrifice all implants were submerged and covered by a healthy ridge of skin. All implants were in situ when animals were euthanized. At retrieval, a macroscopic evaluation of the implant site was performed, which confirmed that all the implants had been correctly inserted, and no signs of infection were observed. Each histological ground section comprised the implant and the surrounding host bone (Figure 5 and Figure 6).

For both types of implants, a favorable bone response was observed on the two implant surfaces; an amount of new bone formation was found in both after 12 weeks of healing. All specimens showed a new bone formation or active bone formation on the inner cutting side of the cortical bone and the inner portion of bone marrow. Both implant surfaces were surrounded by newly formed trabeculae of woven bone (Figure 5 and Figure 6). Histologic evaluation showed that both surfaces are biocompatible and absent of foreign body inflammation, detritus and fibrous tissue around both types of implants.

Histomorphometric analysis revealed that the mean Bone–implant contact (BIC) ratio was similar around the SLA™ surface implants (63.74 ± 10.89) and Nanoblast Plus™ surface implants (62.83 ± 9.91); there was no significant difference between the two types of surface implants (*p* > 0.05; Table 2).

Histological analysis was performed to further characterize the regions of the implants. The bone–implant contact ratio (BIC) was determined on the cervical, medium and apical thirds of the implant The histomorphometric findings obtained from light microscopy showed in both implant surfaces a greater BIC in the medial and apical cervical region and a lower BIC in the cervical region. (Table 3).

## 3. Discussion

Current modifications in the surface of dental implants have been suggested to improve osseointegration. Surface topography and chemical composition are involved in the early stages of wound healing and can improve cell response and tissue attachment to the implant. In fact, these surface modifications can increase the response of the bone tissue around the implant by stimulation of the healing process and formation of new bone [11,12,13,14].

Several studies have shown that a rough implant surface, when compared to a relative smooth surface, promotes better protein adsorption, increases extracellular matrix deposition and improves the differentiation toward osteoblastic cells [23,25,26].

The most widely used commercial technique for the treatment of an implant surface is the combination of blasting and acid-etching. Experimental studies reported that this treated surface increases biocompatibility in the early bone-formation stage and stimulates cell differentiation. After blasting and acid-etching, the rough topography of titanium implant surfaces provides positive effects on the activation of blood platelets and cell migration [22,23,24].

Regarding the implant surface roughness, the cellular viability and osteoblast activity is increased with a roughness between 1 and 100 μm. These roughness and topographical features of the implant surface (peaks and valleys) are important factors in the biological response and the bone–implant interface [27]. The surface roughness of dental implants is usually classified as smooth (Ra < 0.5 μm), minimally rough (Ra 0.5–1.0 μm), moderately rough (Ra 1.0–2.0 μm) and highly rough (Ra > 2.0 μm) [24,25,27].

For this reason, the present research analyzed the surface topography and surface roughness of two types of sandblasted and etched implants with different methods of fabrication. The results of this study showed a greater roughness in SLA™ surface implants than Nanoblast Plus™ surfaces according to Ra, Rq and Rz parameters. The Ra values obtained in the present study were 3.05 μm (SLA™) and 1.78 μm (Nanoblast Plus™). Most commercial dental implants have an Ra value of 1–2 μm. This roughness range seems to be optimal for achieving osseointegration [24,27,28]. In fact, both tested surfaces in the present study showed a similar BIC (63.74 and 62.83%). Several studies have demonstrated that the level of roughness of this implant surface, its chemical composition and topography may affect cell function, adhesion and viability in achieving a higher and faster osseointegration and better biomechanical integrity, resulting in an increase in bone–implant contact (BIC) [29,30]. In fact, recent research showed that the torque removal measurements of implants with sandblasted and etching surface were significantly higher when compared with machined surfaces in the time intervals of 3–6 weeks [31].

The osseointegration and the percentage of bone-to-implant contact (BIC) are highly dependent on the surface properties. Important parameters such as chemical composition can play a crucial role in cellular viability, adhesion and proliferation. Commercially pure titanium showed excellent biological properties [24,27]. After the blasting action, some particles may become embedded and contaminate the implant surface [32,33]. Using acid etching, the most superficial layers of the surface are removed and cleaned. In the results of the present study, titanium appeared as the main element of the surface; however, some residues of the original aluminum particles remained attached to the surface SLA™ (Figure 3) [34]. The presence of organic contamination (carbon and oxygen) on implant surfaces can be explained since the hydrocarbons present in the atmosphere are instantly adsorbed on the titanium surface exposed to the air (Figure 3 and Figure 4) [27]. This limited contamination does not affect the distribution and absorption of proteins and the cellular adhesion in the surface during the early stages of osseointegration [32,34].

Experimental studies in rabbits established that histology and histomorphometry analyses are scientific methods to evaluate the healing of dental implants and to assess the rate and extent of osseointegration [29,35]. An experimental study indicated that the sandblasting and acid-etching surface presented more BIC than the oxidized surface, suggesting that surface could have a stronger affinity for bone than the oxidized surface during the initial healing period [35]. Another study showed that two different sandblasting and acid-etching surfaces (SLA active and a calcium-modified surface) demonstrated a good osteoconduction during the initial 15 days after implant placement, and a higher BIC for both surfaces after 60 days [29].

These results were confirmed during the present study. The implants with a roughened surface by sandblasting and acid-etching had an early fixation in bone tissue and a high percentage of BIC (Figure 5 and Figure 6 and Table 2 and Table 3). Both tested implant surfaces (SLA™ and Nanoblast Plus™) had similar capacities for osseointegration at 12 weeks. The rate of osseointegration of the implant surfaces have been previously analyzed in similar animal models. Histologic and histomorphometric results of this experimental study demonstrate a good bone response with an important formation of new bone around the implant surface after the healing period [29,35].

The macro geometry of the implant is considered as an important factor for osseointegration, and the macrostructure modification may provide a better response [9,36]. However, the results of the present study showed that both macro designs achieved similar percentages of bone anchorage from cervical to apical, indicating that the two surfaces had good biocompatibility and bone conduction [29,35].

The experimental results of this research suggested that sandblasting and acid-etching implant surfaces could be clinically advantageous for shortening the implant healing duration, providing an earlier anchorage, reducing micro motion and thus allowing earlier loading protocols of prosthetic restorations. This may be especially useful on implants placed in areas with low density of bone as the maxilla [2,37].

## 4. Materials and Methods

### 4.1. Implants Characterization

In this study, two different types of implants were used: a control implant (Bone Level SLA™, diameter 4.1 mm, length 10 mm, Straumann, Basel, Switzerland) and a test implant (IPX Nanoblast Plus™, diameter 3.5 mm, length 10 mm, Galimplant, Sarria, Spain). Both implants were manufactured with a sandblasted-etched surface.

To determinate decomposition and structure of the implant surface, characterization was performed using scanning electronic microscopy (SEM, Philips 515, Philips, Eindhoven, The Netherlands) and Energy Dispersive X-ray Spectroscopy (XPS, Thermo Fisher Scientific, Waltham, MA, USA).

Roughness was evaluated in the framework of the recommendations by Wennerberg and Albrektsson [38] on topographic evaluation for dental implants. A white light interferometer microscope (Wyko NT1100, Veeco, Oyster Bay, NY, USA) was used. The surface analysis area was 227 × 298 µm^2^ for both implants. Data analysis was performed using specific software (Wyko Vision 232TM, Veeco, Oyster Bay, NY, USA) using a Gaussian filter to separate waviness and form from the roughness of the surface.

### 4.2. Animal Model

Ten healthy adults New Zealand white rabbits (Granja San Bernardo, Navarra, Spain) of 6–7 months of age and mean weight 5 kg were used, after approval by the ethical committee of the University of Santiago de Compostela (08/14/LU-002). Procedures were conducted in the Animal Experimentation Facility of the University of Santiago de Compostela in Lugo, Spain. All experiments were performed according to the Spanish Government Guide and the European Guide for Animal Care. Animals were housed in enriched rabbit cages, allowing normal activity, and were monitored once a day by trained staff to assess changes in general health. Animals had free access to food and tap water. The implants were evaluated according to UNE-EN ISO 10993-6 following protocols previously published [35].

All surgical procedures were performed in an operating room under sterile conditions, and under general anesthesia induced and maintained on a concentration of 2.5–4% of isoflurane (Isoba-vet, Schering-Plough, Madrid, Spain). The animals were first sedated with a combination of medetomidine (50 mg/kg/i.m., Domtor, Esteve, Barcelona, Spain) and ketamine (25 mg/kg/i.m., Imalgène 1000, Merial, Toulouse, France). During anesthesia, the animals were continuously monitored by a veterinarian category B or C.

Animals received antibiotic prophylaxis for one week with enrofloxacin (15 mg/Kg/s.c. once a day, Ganadexil 5%, Invesa, Barcelona, Spain) and pain was controlled with meloxicam (0.2 mg/Kg/s.c., Metacam, Boehringer Ingelheim, Barcelona, Spain) for three days.

Each animal received two implants (one control and one test implant) in the right and left femur, for a total of twenty implants in 10 animals. The experimental site was located on both distal lateral condyles of the femur. After three weeks of quarantine, the animals were induced with general anesthesia. After shaving and disinfecting, the femoral condyles were exposed by a lateral longitudinal incision. Implant bed preparations were carried out according to the recommendations of the manufacturers. Finally, the muscle, the subcutaneous tissue and the skin were sutured in layers with resorbable sutures (Vicryl 4-0, Ethicon, Raritan, NJ, USA).

A total of 12 weeks later, the rabbits were painlessly sacrificed by a sodium pentobarbital overdose (100 mg/Kg/i.v., Dolethal, Vétoquinol, Madrid, Spain) after sedation with ketamine and medetomidine.

### 4.3. Histological and Histomorphometric Analyses

The blocks containing the implant and the distal distracted femur bone with hard and soft tissues around the implant were obtained using an oscillating saw, fixed and identified. These blocks were dehydrated in different graded ethanol series (70–100%) and infiltrated with different graded mixtures of ethanol and glycometacrylate (Technovit 7200 VLC, Heraus Kulzer, Werheim, Germany) following guidelines previously published [39]. The samples were then polymerized and heated at 37 °C for 24 h to assure complete polymerization.

Longitudinal sections (central sections) in the direction of 200 microns were carried out on implants using a band saw and mechanically micropolished (Exakt Apparatebau, Norderstedt, Germany) using 1200 and 4000 grit silicon carbide papers (Struers, Copenhagen, Denmark) until we obtained samples with a thickness of approximately 40 μm. The slides were stained using the Levai–Laczkó method [40] for both histological and histomorphometric analysis.

Quantitative and semiquantitative histology was performed using a motorized light microscopy and a digital camera connected to a PC-based image capture system (BX51, DP71, Olympus Corporation, Tokyo, Japan). Peri-implant tissues were the regions of interest. Image analysis was conducted based on color and shape of these structures, differentiating the new and lamellar bone from the connective and vascular tissues (Adobe Photoshop, San Jose, CA, USA). Parameters were evaluated and measured by a masked examiner using the PC-based image analysis program CellSens 1.5 (Olympus Corporation, Tokyo, Japan).

Firstly, a semiquantitative histological evaluation was performed according to the ISO 10993-6 standards. The following grading scale was used: absent, 1 = slight, 2 = moderate, 3 = marked and 4 = severely irritant.

The implant surface in contact with mineralized bone, referred to as “bone to implant contact” (BIC) was calculated as a percentage on the cervical, medium and apical thirds of the implant (Figure 7).

### 4.4. Statistical Analysis

The animal was chosen as the unit for the statistical analysis. The primary outcome parameter was BIC at the different compartments. The data were reported by using means, standard deviations (SD), ranges, 95% confidence intervals (CI) and medians (SPSS, SPSS Inc., Chicago, IL, USA). A paired two-sample t test was performed. Results were considered as significant at *p* < 0.05.

## 5. Conclusions

The results of this study suggested that both implant surfaces providing a histologic fixation after 12 weeks of healing in the femoral bone of an experimental rabbit model. Histologic and histomorphometric analysis of the bone-implant contact indicated a good osseointegration of two tested implants. Both sandblasted and acid-etched implant surfaces show an intimate interaction with newly formed bone, indicating an excellent biologic response.

## Figures and Tables

**Figure 1 ijms-22-08507-f001:**
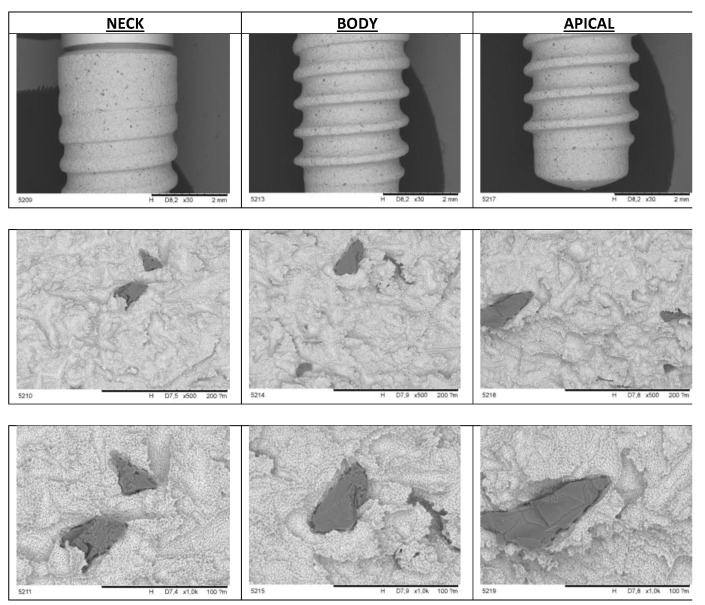
Scanning Electronic Microscope images of the SLA™ surface.

**Figure 2 ijms-22-08507-f002:**
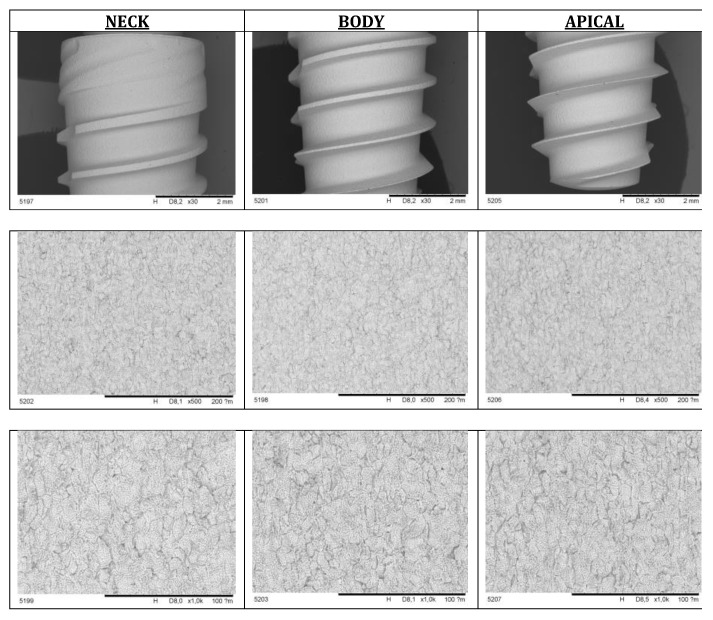
Scanning Electronic Microscope images of the Nanoblast™ surface.

**Figure 3 ijms-22-08507-f003:**
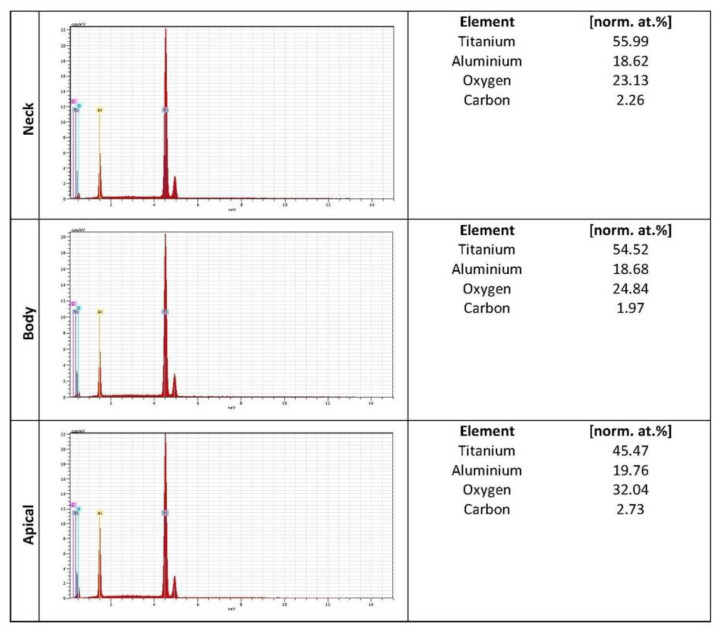
Chemical composition of SLA™ surface.

**Figure 4 ijms-22-08507-f004:**
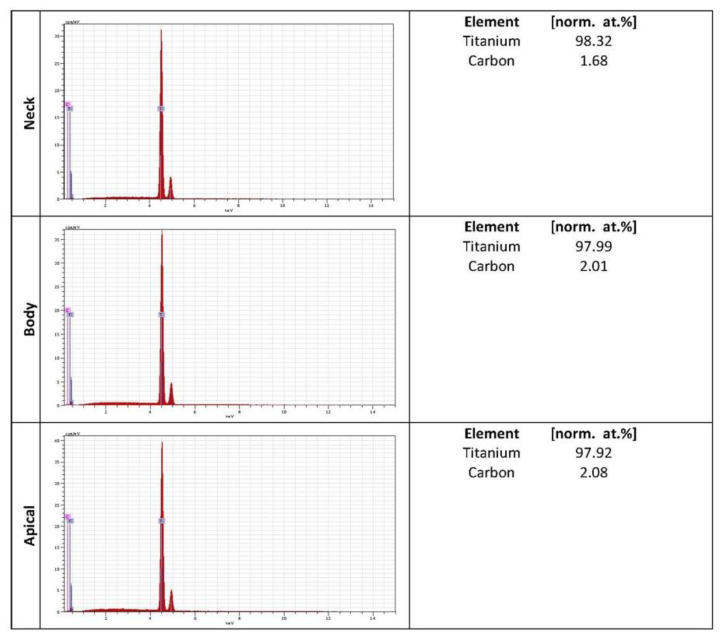
Chemical composition of Nanoblast Plus™ surface.

**Figure 5 ijms-22-08507-f005:**
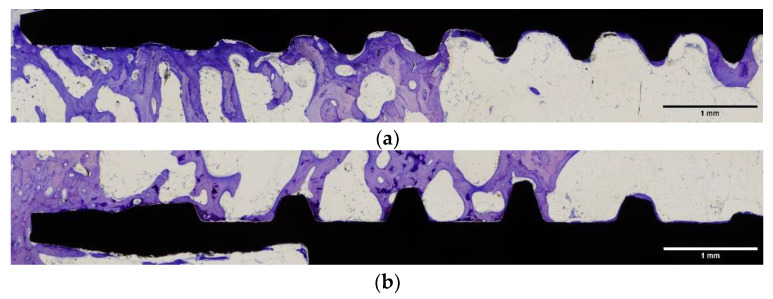
Amplified light microscope images of the two surface implants tested: (**a**) SLA™ surface implant; (**b**) Nanoblast Plus™ surface implant. Bars show magnification.

**Figure 6 ijms-22-08507-f006:**
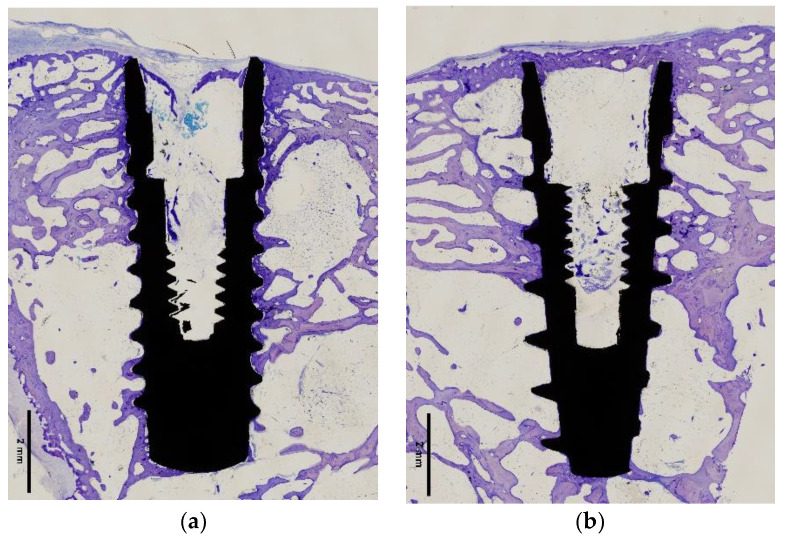
Light microscope images of the two surface implants tested: (**a**) SLA™ surface implant; (**b**) Nanoblast Plus™ surface implant. Bars show magnification.

**Figure 7 ijms-22-08507-f007:**
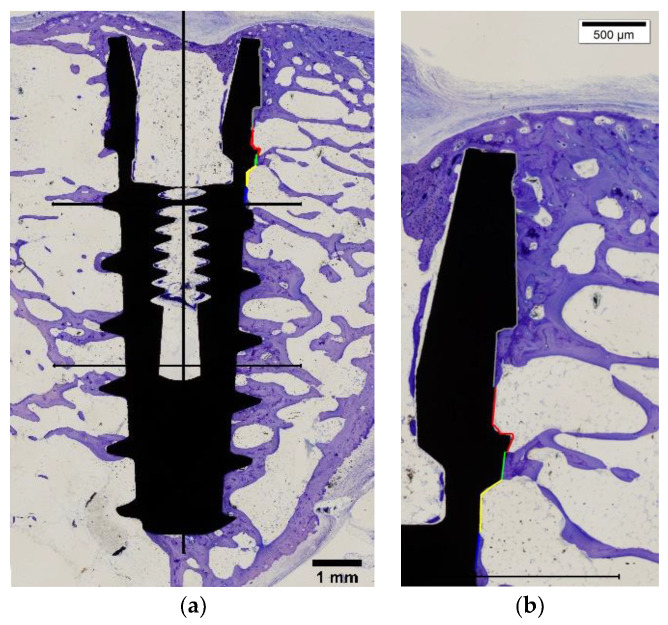
Light microscope image. (**a**) Scheme of distribution of BIC measurements on the cervical, medium and apical thirds of the implant. (**b**) View of cervical region with the measurements. Bars show magnification.

**Table 1 ijms-22-08507-t001:** Surface profile parameters (μm).

Parameter	SLA™	Nanoblast Plus™
Ra	3.05	1.78
Rq	3.81	2.23
Rz	21.50	14.11
*p*	>0.05	

**Table 2 ijms-22-08507-t002:** % Bone–implant contact.

Animal	SLA™	Nanoblast Plus™
1	48.80	53.68
3	55.96	68.82
4	60.01	71.52
5	57.65	71.70
6	67.35	70.77
7	67.28	52.40
8	94.41	69.28
9	51.96	61.72
10	70.22	45.57
Mean ± SD (median)	63.74 ± 13.61 (60.01)	62.83 ± 9.91 (68.82)
*p*	0.910	

**Table 3 ijms-22-08507-t003:** Regional distribution of BIC.

Region	SLA™	Nanoblast Plus™	*p*
Cervical	20.14 ± 4.27	18.78 ± 4.61	0.55
Medial	22.39 ± 5.49	21.35 ± 4.84	0.67
Apical	21.21 ± 7.14	22.69 ± 5.26	0.60
Total	63.74 ± 13.61	62.83 ± 9.91	0.91

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
