# Peer review of "Osseointegration of Sandblasted and Acid-Etched Implant Surfaces. A Histological and Histomorphometric Study in the Rabbit"

_ijms, 2021, doi:10.3390/ijms22168507_

Round 1
Reviewer 1 Report
The experimental design is hard to follow, especially the implant design with grooves and spiral fins. The roughness of the implant surface cannot be evenly distributed, especially the roughness produced by the SLA method.
In addition, the nanoblast has micrometer roughness, the nomenclature is strange and the major SLA processes were missing in materials and methods.
The roughness designs of Ra 3.05 and 1.78 um is too close to confirm the verification result.
Author Response
Dear reviewer:
Firstly thank you for your comments. I will try to reply to all your suggestions.
- The experimental design is hard to follow, especially the implant design with grooves and spiral fins. The roughness of the implant surface cannot be evenly distributed, especially the roughness produced by the SLA method.
- In addition, the nanoblast has micrometer roughness, the nomenclature is strange and the major SLA processes were missing in materials and methods.
Dear reviewer. We have tried to explain better the experimental design and results according your recommendations. I hope that now are according to your requirements.
- The roughness designs of Ra 3.05 and 1.78 um is too close to confirm the verification result.
Dear reviewer. Some authors preclude that roughness of the implants may play a role on the osseointegration. Ra of 3.05 (SLA) is considered as a highly rough implant. However a Ra of 1.78 would be considered as a moderately rough implant. However we did not found differences in terms of osseointegration, but literature classified them in different groups.
Reviewer 2 Report
Overall a well-written article, but the following suggestions should be improved:
Introduction)
There have already been many histological and histomorphometric studies comparing SLA and sandblasted implants. The advantages of each surface treatment were well described in paragraphs, but it seems that the reason why the research was initiated clearly, such as why the two products were compared, or what was lacking in the research so far, need to be described in the last paragraph of the introduction.
Results)
In Table 2, “Bone to Implant Contact” results were described by confusing the decimal points and commas.
To clearly distinguish the thirds cervical, medial and apical of the implant in Figure 7, it will be easier for readers to understand if an enlarged figure is added to the right side.
Table 1 and 2 does not include statistical results. It is only described in the text. Text and picture must match.
It is recommended to check the magnifications of Figure 5 and 6, and display the same size if the magnification is the same.
Discussion)
In the discussion, the contents of the introduction are repeated excessively. An appropriate introduction is fine, but it is recommended to write it mainly for discussion of the results.
Avoid using "Probably" in 221 when writing an article.
Author Response
Dear reviewer:
Firstly thank you for your comments. I have tried to modify the text with all your suggestions. In addition we have review the language. If you consider that is not enough we will send again to an English reviewer.
Overall a well-written article, but the following suggestions should be improved:
Introduction)
There have already been many histological and histomorphometric studies comparing SLA and sandblasted implants. The advantages of each surface treatment were well described in paragraphs, but it seems that the reason why the research was initiated clearly, such as why the two products were compared, or what was lacking in the research so far, need to be described in the last paragraph of the introduction.
Dear reviewer. We have improved the last paragraph to explain the reasons of the research.
Results)
In Table 2, “Bone to Implant Contact” results were described by confusing the decimal points and commas.
Thank you for your observations. We have changes the commas for points. We believe that now are corrected.
To clearly distinguish the thirds cervical, medial and apical of the implant in Figure 7, it will be easier for readers to understand if an enlarged figure is added to the right side.
We have included a magnified figure in the right side.
Table 1 and 2 does not include statistical results. It is only described in the text. Text and picture must match.
We have modified the tables according to your recommendations.
It is recommended to check the magnifications of Figure 5 and 6, and display the same size if the magnification is the same.
We have checked the magnifications of the images. All images were captured with the same x4 objective, but some were amplified to show details. The bars always showed the magnification. We have included it in the figures text.
Discussion)
In the discussion, the contents of the introduction are repeated excessively. An appropriate introduction is fine, but it is recommended to write it mainly for discussion of the results.
We have shorted the introduction and discussion to avoid the repetition. Thank you for your suggestions.
Avoid using "Probably" in 221 when writing an article.
I apologize for my mistake. We have modified the paragraph.
Round 2
Reviewer 1 Report
Since the author has tried to modify, but strongly recommends that the author improve the research design in future investigations, especially the control group and parameter control, I think it is acceptable.